# EXOVIP: STEP-BY-STEP VERIFICATION AND EXPLORATION WITH EXOSKELETON MODULES FOR COMPOSITIONAL VISUAL REASONING

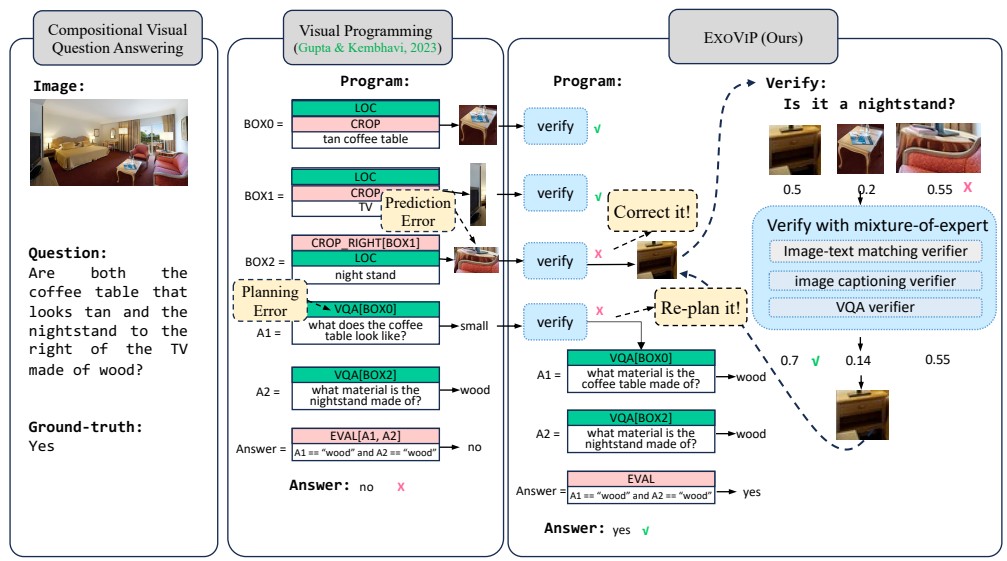

Figure 1: An overview of EXOVIP. The prediction after each step is verified by the proposed "Exoskeleton" verification modules, which contain a mix of three sub-verifiers. The verified scores help correct the errors in the vision module predictions or refine the reasoning programs planned by LLM.

## ABSTRACT

Compositional visual reasoning methods, which translate a complex query into a structured composition of feasible visual tasks, have exhibited a strong potential in complicated multimodal tasks like visual question answering, language-guided image editing, etc. Empowered by recent advances in large language models (LLMs), this multimodal challenge has been brought to a new stage by treating LLMs as few-shot/zero-shot planners, *i.e.*, visual-language programming (Gupta & Kembhavi, 2023). Such methods, despite their numerous merits, suffer from challenges due to LLM planning mistakes or inaccuracy of visual execution modules, lagging behind the non-compositional models. In this work, we devise a "plug-and-play" method, EXOVIP, to correct the errors at both the planning and execution stages through introspective verification. We employ verification modules as "exoskeletons" to enhance current vision-language programming schemes. Specifically, our proposed verification module utilizes a mixture of three sub-verifiers to validate predictions after each reasoning step, subsequently calibrating the visual module predictions and refining the reasoning trace planned by LLMs. Experimental results on two representative vision-language programming methods showcase consistent improvements on five compositional reasoning tasks on standard benchmarks. In light of this, we believe EXOVIP can foster better performance and generalization on open-domain multimodal challenges.

## 1 INTRODUCTION

Compositional visual reasoning tasks, such as visual question answering or image editing following language instructions, are challenging multimodal tasks that require complex multi-step visual reasoning based on the language instruction. Compositional methods like neural modular networks (Andreas et al., 2015; Hu et al., 2017; Johnson et al., 2017; Hu et al., 2018; Le et al., 2022; Qian et al., 2022), which translate the complex language instruction into feasible individual visual tasks, has been successful in this task. However, traditional compositional methods require well-designed neural modules for specific datasets, thus struggle in generalization to open domains. In addition, the intermedia embedding and attention among the neural modules can not be improved by introducing supervision signals or feedback, so the performance of these works is limited to the end-to-end training mechanism. Recently, empowered by the advances in large language models (LLMs) such as in-context learning and train-of-thought reasoning (Radford & Narasimhan, 2018; Radford et al., 2019; Brown et al., 2020; OpenAI, 2023; Chowdhery et al., 2022). recent methods like VISPROG (Gupta & Kembhavi, 2023) and ViperGPT (Dídac et al., 2023) apply LLMs as zero-shot/few-shot planners to solve visual reasoning tasks, *i.e.* visual language programming. These visual language programming methods leverage off-the-shelf pretrained vision models and compose them step by step according to the reasoning trace planned by LLMs, yielding interpretable intermediate results and highly generalizable reasoning ability.

However, despite their merits, current visual programming methods still suffer from challenges due to the failure of the LLM planning or the visual modules, lagging behind the performance of non-compositional models. To analyze the drawbacks, we manually checked 100 randomly sampled failure cases of VISPROG (Gupta & Kembhavi, 2023) on the visual question answering GQA dataset (Hudson & Manning, 2019). We find that most of the failures can be classified into two categories: (1) around 30% of the failures are due to planning errors: LLM can not parse the language query into a correct solvable program; (2) more than 40% of the failures are due to module error: the visual modules are not able to correctly execute the program. The others (less than 30%) are caused by synonyms (*e.g.* "woman" vs "lady") or ambiguity in the questions. More details, including statistics and examples of the failure cases, can be found in Appendix C.

Motivated by these failure modes, in this work, we introduce EXOVIP, a "plug-and-play" method that uses "exoskeleton" verification modules to verify the reasoning results step by step, thus correcting the module errors and refining the LLM planning traces. In Fig. 1, we demonstrate how EXOVIP helps correct the two types of errors. Specifically, the verification module contains a mixture of three sub-verifiers, including an image-text matching verifier, an image captioning verifier, and a visual question answering verifier. The verification module validates the correctness of the predictions of the vision modules step by step and calibrates them to correct the module errors. Furthermore, to refine the planning traces, we build a reasoning trace tree based on the verification scores as well as the self-correctness score from LLMs (Pan et al., 2023), and search through the tree to find the best trace that has the highest score.

To demonstrate the effectiveness of EXOVIP, we apply our method to two recent visual programming methods: self-defined programs, *i.e.*, VISPROG (Gupta & Kembhavi, 2023) and Python code programs, *i.e.*, ViperGPT (Dídac et al., 2023). We run experiments on five compositional visual reasoning tasks: compositional image question answering on GQA Hudson & Manning (2019); referring expression understanding on RefCOCO and RefCOCO+ (Yu et al., 2016; Kazemzadeh et al., 2014), natural language for visual reasoning on NLVR (Suhr et al., 2019), visual abstract reasoning on KILOGRAM (Ji et al., 2022), and language-guided image editing on MagicBrush (Zhang et al., 2023a). Experiment results show consistent improvements with the two models on the five tasks. In light of this, we believe EXOVIP can foster better performance on open-world compositional reasoning tasks. To summarize, our main contributions are as follows:

- We introduce the "exoskeleton" verification modules for compositional visual reasoning, which verifies the correctness of vision module predictions step by step.
- We show how the verification modules are leveraged to correct the module errors by calibrating the module predictions, and to correct the planning errors by tree searching considering both verification scores and LLM self-correctness.
- We apply our method on two models and show consistent improvements over five tasks, showing the effectiveness of EXOVIP.

## 2 RELATED WORK

**LLMs in multimodal tasks.** LLMs brought great convenience to multimodal tasks with their generalizability and knowledgeability. Generally, there are three ways researchers use LLMs to solve multimodal tasks. Some researchers incorporate additional parameters to adjust LLMs for use in multimodal domains, then fine-tune the model with the LLMs either frozen (Tsimpoukelli et al., 2021; Alayrac et al., 2022; Li et al., 2023b; Gao et al., 2023; Li et al., 2023a; Dai et al., 2023; Zhang et al., 2023d) or unfrozen (Hao et al., 2022; Huang et al., 2023; Peng et al., 2023). Others take language model as an expert, and mixture it with experts from other modalities, such as vision, speech to collaborate on various kinds of multimodal tasks (Zeng et al., 2023; Zhang et al., 2023c; Liu et al., 2023b). In this work, we mainly focus on the third way which adopts LLM's planning ability in parsing complex queries. VISPROG (Gupta & Kembhavi, 2023) takes LLM to compose models for queries by generating programs. The strong zero-shot performance of VISPROG on a range of vision-language tasks demonstrates its potential in multimodal tasks involving complex reasoning. ViperGPT (Dídac et al., 2023) leverages LLMs to generate Python code, which composes a set of available modules. MM-REACT (Yang et al., 2023) builds a multi-round, dialogue-based system to call a set of vision experts by designing the prompt of LLMs. However, the performances of these works are hindered by both the parsed planning chain and the visual experts. Inspired by the excellent performance gain from the step-by-step verification (Lightman et al., 2023), we improve this train of work with additional verification strategies.

**Compositional multimodal methods.** Compositional methods have long been explored to improve neural models' interpretability and reasoning ability. At an early stage, neural module networks (NMN) (Andreas et al., 2015; Hu et al., 2017; Johnson et al., 2017; Hu et al., 2018; Le et al., 2022; Qian et al., 2022) compose neural models to end-to-end differentiable networks. However, the pre-defined neural modules have limited applications on open-domain challenges, and the intermedia embedding and attention makes it difficult to construct intermedia supervision signals. Recently, the presence of LLMs has made it possible to automatically compose various kinds of finetuned neural models (Zeng et al., 2023; Gupta & Kembhavi, 2023; Dídac et al., 2023; Yang et al., 2023; Liu et al., 2023b) or external tools (Parisi et al., 2022; Khot et al., 2023; Schick et al., 2023; Shen et al., 2023; Lu et al., 2023; Qin et al., 2023). These works allow us to diagnose the intermedia rationales of the reasoning process. However, human annotation of these intermedia results can be rather time-consuming. In this work, we make ways to correct errors in the intermedia results without any human intervention.

**Self-correctness in LLMs.** Although LLMs achieve great success in various tasks, there are many errors in LLM-based natural systems (Pan et al., 2023): hallucination (Li et al., 2023c; Zhang et al., 2023b), unfaithful reasoning (Golovneva et al., 2022; Ribeiro et al., 2023; LYU et al., 2023), toxic, biased, and harmful contents (Shaikh et al., 2022), flawed code. One popular way to fix these errors is to use the LLMs themselves (Madaan et al., 2023; Shinn et al., 2023; Ye et al., 2023; Yan et al., 2023) to obtain feedback, which can be adopted to correct the errors. Motivated by the self-correction capability of LLMs in addressing mistakes from LLM-powered natural language systems, some researchers introduce the self-correcting strategy to reduce the reasoning chain in multimodal frameworks. IPVR (Chen et al., 2023) additionally utilizes LLMs to generate the rationale supporting the answer, checks the generated rationale with a cross-modality classifier, and makes sure that the rationale can consistently infer the predicted output. IdeaGPT (You et al., 2023) takes another LLM as a reasoner to get the final answer by summarizing the intermedia results from visual experts. Additionally, the reasoner helps to improve the results iteratively through self-consistency. However, it's intuitive that LLM's self-correction ability would be limited by the LLM itself. In our work, we combine the feedback from LLM and other visual experts to verify the intermedia results and the planned reasoning chain.

## 3 PRELIMINARIES

**Task Definition.** Our work focuses on a set of Visual Compositional Reasoning (VCR) tasks, such as visual question answering, referring expression understanding, visual reasoning using natural language, abstract reasoning, language-guided image editing. These VCR tasks require compositional reasoning about an image input $I$ and a text input $T$, and predict the output, *e.g.* answer to a given question, edited images given a language instruction, etc.

**Visual-Language Programming (VISPROG).** VISPROG (Gupta & Kembhavi, 2023) is a zero-shot model for the VCR tasks, utilizing LLMs and pretrained vision models. VISPROG first uses LLMs to decompose the complex text description into a sequence of individual operations, then executes each operation by calling various pretrained visual operation models, including object detectors, image captioners, VQA models, image generators, *etc*. In other words, different vision models are composed in a way that is specified by the LLM to get the prediction. Given the input text $T$, an LLM transforms it into an executable program $P$ containing a sequence of operations: $P = \{o^1, \ldots, o^n\}$, where $n$ is the number of operations. Each operation $o^i$ can be executed by some symbolic operations (*e.g.*, "crop", "and", "or"), or by calling some pretrained visual models (*e.g.* CLIP (Radford et al., 2021), BLIP (Li et al., 2022b)). The output of operation $o^i$ is denoted as $a_i$. The final prediction is derived after we execute all the operations. However, this perspective highlights two key shortcomings of existing approaches: i) module error, the operation models can not predict the answer correctly; ii) planning error, the LLM might generate unfaithful reasoning.

# 4  EXOVIP: EXOSKELETONS WITH VERIFICATION AND EXPLORATION

To address the aforementioned shortcomings, we propose EXOVIP, a framework that adopts exoskeleton verification modules to calibrate the prediction of the execution modules and refine the reasoning path with tree searching. Fig. 1 depicts the overall framework.

For each operation $o^i$, we get a set of candidate answers $\{a_1^i, \ldots, a_k^i\}$, with confidence scores $\{p_1^i, \ldots, p_k^i\}$. Unlike VISPROG, which directly takes the top answer, we use additional verification modules to verify each candidate answer, thus producing verification scores $\{s_1^i, \ldots, s_k^i\}$. Then the verification scores $s$ are used to calibrate the original scores, so the errors made by the execution modules can be corrected. Additionally, we use the verification scores to search for a program with high verification scores, in order to refine the execution program $P$ by tree-searching.

In this section, we will first introduce the verification modules, and then describe how the verification results are applied to correct the results of execution modules, and to search for the reasoning trace.

## 4.1  VERIFICATION MODULES

The verification modules aims to verify the candidate answers $\{a_1^i, \ldots, a_k^i\}$ given an operation $o^i$. For example, the `LOC(nightstand)` operation returns a set of candidate bounding boxes containing a nightstand, then the verification module verifies whether each of the returned boxes contains a nightstand and produces verification scores.

Our verification module is a mixture of three sub-verifiers, including an image-text matching verifier, an image captioning verifier, and a visual question answering verifier. Each verifier is a pretrained vision-and-language model that is taken off the shelf. The outputs of the three verifiers are combined as the final verification score. Note the verification model does not introduce additional pretrained models, as these verifiers are from the execution modules of VISPROG.

**Image-text matching verifier** calculates the similarity between the whole images and all candidate sentences, which returns the semantic representation of the image-sentence pair. We construct the candidate sentences $\mathcal{T}_{ans}$ by filling the template "a photo of" with candidate answers. In this work, we select CLIP (Radford et al., 2021) to calculate the similarity between images and sentences.

$$s_{ans}^{itm} = \text{ITM}(\mathcal{T}_{ans}, img) \tag{1}$$

**Image captioning verifier** leverages natural language to describe the visual details of the image. We first get the caption of the image $\mathcal{C}_{img}$ by BLIP (Li et al., 2022b). We then construct the descriptions of candidate answers $\mathcal{C}_{ans}$ with the template "the image describe". Specifically, for candidate question-answer pairs, we initially transform the pair into a sentence before inserting it into the template. After that, we calculate the sentence semantic similarity (Reimers & Gurevych, 2019) between the captions and the constructed descriptions as the verification score.

$$s_{ans}^{cap} = Sim(\mathcal{C}_{ans}, \mathcal{C}_{img}) \tag{2}$$

**Visual question-answering (VQA) verifier** is more flexible than others, which offers us more opportunities to evaluate the advanced relationships between image and language, such as entailment

and factual consistency. Slightly different from the other two types of models, for VQA verifier, we design templates w.r.t. the neural modules. For example, we use "Is there any object in the image ?" for the object detection model, and use "Does this part looks like object ?" for the classification model used in the abstract reasoning task. We determine the verification score by BLIP (Li et al., 2022b) by calculating the difference in answer probabilities $\mathcal{Q}_{ans}$ between "yes" and "no".

$$s_{ans}^{vqa} = \text{VQA}(\mathcal{Q}_{ans}, True) - \text{VQA}(\mathcal{Q}_{ans}, False) \tag{3}$$

**Verification score**   Having the scores from each individual verification module, we compute their average to get the verification score for each given answer.

$$s_{ans} = \text{avg}(s_{ans}^{itm}, s_{ans}^{cap}, s_{ans}^{vqa}) \tag{4}$$

**Negative sampling.** Empirically, we find that directly applying this verification score does not work well, because the score scales for different kinds of candidates are not well-calibrated. Motivated by recent works in truthfulness (Li et al., 2022a), commonsense (Ye et al., 2022), and bias (Ruggeri & Nozza, 2023), we propose to take the difference of a candidate answer $a_j$ with its antonym $n_j$ as the final verification score. More specifically, the antonym $n_j$ is selected based on the text embeddings from CLIP Radford et al. (2021), *i.e.* the word of lowest embedding similarity is selected. For example, the antonym of "nightstand" is "stocking". We then compute the difference of the verification scores of the candidate answer and its antonym, and get the final verification score. Mathematically, given a candidate answer $a^j$, the final verification score is

$$s_j = s_{a_j} - s_{n_j} \tag{5}$$

**Calibration using verification scores**   After obtaining the verification scores of all candidate answers $S = \{s_1, \ldots, s_k\}$, we normalize them as weights and calibrate the candidate predictions.

$$p'_j = w_j * p_j, \tag{6}$$

where $w_j$ is the normalized verification score. More specifically, the verification score $s_j$ is re-scaled to $w_j = \frac{s_j - s_{min}}{s_{max} - s_{min}} \cdot (\tau - \frac{1}{\tau}) + \frac{1}{\tau}$, where $\tau$ is a hyper-parameter controlling the scaling factor ($s_{min}, s_{max}$ are the minimum or maximum of all the candidate scores.

## 4.2 EXPLORATION WITH REASONING TRACE

To correct the second type of reasoning errors, *i.e.* planning errors, we further apply the verification scores to refine the reasoning trace predicted by LLMs. Motivated by the recent works showing that searching through a combinatorial problem space can greatly improve the performance of LLMs for complex tasks (Yao et al., 2023; Khalifa et al., 2023; Hao et al., 2023), we introduce our dynamic reasoning trace searching procedure, which takes advantage of both the LLM self-correctness potential and our verification modules.

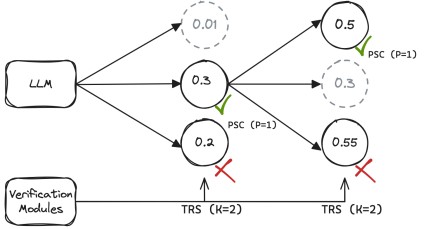

Figure 2: Search of the reasoning trace. We ~~beam~~ search through the program tree, based on the verification scores as well as the LLM self-correctness.

**Tree-based reasoning trace searching (TRS)** The reasoning trace searching procedure is represented as a tree structure, where each node of the tree is a reasoning operation. To get the best reasoning trace, we search from the tree using the beam search algorithm (Graves, 2012; Boulanger-Lewandowski et al., 2013; Sutskever et al., 2014), which has long been proven to be effective in sequence-to-sequence problems. In each step of searching, we consider both the verification scores and the LLM self-correctness scores. More specifically, our trace searching procedure contains two ~~three~~ steps. First, in order to generate more diverse reasoning traces to search from, we randomly perturb the in-context examples (*i.e.* change the order or remove some samples of examples) in the prompt for LLM. Second, after we get the result of candidate neural modules, we sort them according to the accumulative verification scores and select the top $K$ candidate reasoning traces.

**Post-hoc self-correction (PSC)** ~~Third, b~~Because the verification scores can be very close for the selected $K$ traces, we further use the self-correctness ability of LLMs to reorder the $K$ traces and select the top $P$ from them ($P < K$). More details of the prompts used for LLM self-correction are included in Sec. E.2. If the verification score is zero at some step, we re-plan the search trace.

## 5 EXPERIMENTS

We set up experiments on the following five tasks. Refer to Appendix E for implementation details.

### 5.1 SETUP

We set up experiments on the following five tasks. Refer to Appendix E for implementation details.

**Compositional image question answering on GQA.** GQA (Hudson & Manning, 2019) is a large-scale dataset containing complex reasoning questions about real-world images in MSCOCO-style (Lin et al., 2014). Considering the large size of the dataset, in order to balance the cost of LLM API and the diversity of evaluation dataset, we follow the setting of VISPROG Gupta & Kembhavi (2023) and sample a subset from GQA for evaluation. We randomly sample 5 samples from the balanced val set and 20 samples from testdev set of each question type. *e.g.* "weatherVerify" for judging the weather, "twoCmomon" for judging common attributions of two objects. In summary, there are 102 question types and 2327 questions in our test set.

**Referring expression understanding on RefCOCO and RefCOCO+.** Given a natural language query describing a region in a given image, the referring expression understanding task requires identifying the bounding box of the object in the image being referred to. RefCOCO and RefCOCO+ (Yu et al., 2016; Kazemzadeh et al., 2014) are two standard datasets for this task. We randomly sample 2 samples per type from the test set from RefCOCO dataset and RefCOCO+ dataset. In summary, our test set includes 66 types, *e.g.* "bicycle", "backpack", and 261 queries.

**Natural language for visual reasoning on NLVR2.** In NLVR2 (Suhr et al., 2019), given a description of a collection of images, the model needs to justify whether the description is correct or not (binary classification). The task requires dealing with various kinds of linguistic phenomena, like numerical expressions, quantifiers, coreference, negation, etc. In this work, we use the NLVR2 balanced test set for evaluation, which includes 2316 questions and corresponding image pairs.

**Visual abstract reasoning on KILOGRAM.** KILOGRAM (Ji et al., 2022) contains richly annotated tangram puzzles and requires the model to understand the abstract tangram shapes (*e.g.* dog, bird) and classify them. Specifically, given a textual description and a set of images, the task is to select the image corresponding to the description. This task evaluates the ability to generalize through abstraction, using visually ambiguous stimuli. We conduct experiments using the test set, where the textual descriptions solely contain the whole-shape description, and the images include parts with different colors. The test set contains 1,251 descriptions, with each one paired with 10 images.

**Language-guided image editing on MagicBrush.** This task requires editing an image according to a natural language instruction, keeping the other area of the image unrelated to the instruction unchanged. The MagicBrush dataset (Zhang et al., 2023a) supports various editing scenarios including single-/multi-turn. Considering the accuracy of automatic evaluation metrics and the costs of human evaluation, in our experiments, we only choose the samples involving single-turn image editing to evaluate our method. In total, there are 100 examples in the test set. Following (Zhang et al., 2023a), we select the CLIP-I and DINO, which measure the image quality with the cosine similarity between the generated image and reference ground truth image using their CLIP (Radford et al., 2021) and DINO (Caron et al., 2021) embeddings.

### 5.2 MAIN RESULTS

We first apply EXOVIP to VISPROG and show results on the five tasks. Then we apply it to the python-code-based compositional reasoning method ViperGPT to demonstrate its generalizability.

#### 5.2.1 COMPOSITIONAL VISUAL QUESTION ANSWERING

**Baseline Model** We set up the experiments following the settings in the official VISPROG implementation.[1] Moreover, we select BLIP-flant5-xxl(Li et al., 2023b) and InstructBLIP-flan-t5-xl(Dai et al., 2023) as additional baselines, which are strong vision-language models incorporating LLMs and pretrained on large vision-language datasets. These baselines have shown strong zero-shot ability on various tasks like image caption and visual question answering.

---

[1]Becasue VISPROG doesn't release their sampled evaluation subset, we do sampling following the VISPROG paper and evaluate all the methods on our sampled evaluation set.

Table 1: Results of compositional visual question answering on GQA. Llava-1.5-13b* is tuned on GQA training corpora, and evaluated with additional prompt.

|   | Methods | Accuracy |
|---|---|---|
|   | BLIP2-xxl (Li et al., 2023b) | 49.20 |
|   | InstructBLIP-flant5-xl (Dai et al., 2023) | 55.39 |
|   | Llava-1.5-13b* (Liu et al., 2023a) | 74.56 |
| 0 | VISPROG (Gupta & Kembhavi, 2023) | 57.41 |
| 1 | EXOVIP w/o self-correctness & negative sampling & search | 57.11 |
| 2 | EXOVIP w/o self-correctness & search | 58.53 |
| 3 | EXOVIP w/o self-correctness (TRS) | 60.57 |
| 4 | EXOVIP w/o verification (PSC) | 60.16 |
| 5 | EXOVIP | 61.49 |

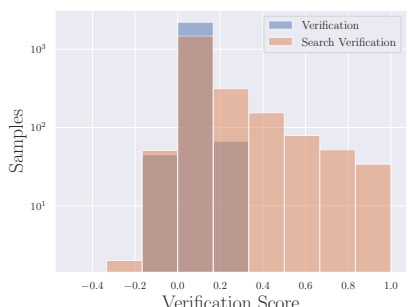

Figure 3: Distribution of verification scores w. and w/o trace searching.

**Analysis** We apply our method to VISPROG and report the results on GQA in table 1. While VISPROG has already demonstrated good performance (57.41) compared with BLIP2 and InstructBLIP, our method further improves its performance to 61.49, showing a significant performance boost. Note that our method does not introduce extra modules or knowledge compared with VISPROG, since the verification modules come from VISPROG itself.

To verify the effectiveness of each component in our method, we run a series of analysis experiments on our method (also in Tab. 1). We have the following observations:

Table 2: Analysis on the sub-verifiers.

| Methods | Accuracy |
|---|---|
| Base | 58.14 |
| ITM | 59.26 |
| Caption | 59.22 |
| VQA | 59.35 |
| All | 60.03 |

a. *Negative sampling is key to verification modules.* Naively adding the verification modules (line-1) does not work, even making the performance worse. But when we introduce the negative sampling strategy using antonyms to the verification modules (line-2), the performance boost becomes significant.

b. *Exploration with reasoning trace matters.* In line-3, "whole search" means we use LLMs to obtain a set of complete planning traces, then execute all the traces to get the final verification scores, and select the best trace with the highest verification score. The "beam search" strategy (line-4) means we select next step according to current verification scores. While "whole search" helps, "beam search" can further improve the accuracy to 60.57 from 59.17, which indicates the effectiveness of our tree-like step-by-step searching strategy.

c. *Self-correctness does help but is less significant than verification mechanism.* In line-5, We only use LLM self-correctness during trace searching, without using the verification scores. While the result shows an accuracy gain of 2.75 over the original VISPROG, applying both leads to further better performance.

**Analysis on the sub-verifers.** We evaluate the effects of different types of verification modules with the setting of the best demonstration setting. As is illustrated in Table 2, Different verification modules share similar boost gain, but a mixture of these modules can benefit more.

**Analysis on the trace-searching strategy.** We calculate the verification scores among different samples and plot the distribution of the verification scores in Tab. 2. We find two advances brought by the searching strategy. First, the average of the verification scores significantly improved after we applied our search strategy. Secondly, the variance gets larger after applying the search strategy, which indicates our method can potentially make use of the verification scores to prompt the effectiveness of the reasoning traces.

**Analysis on invalid programs.** We calculate the percentage of failure cases that can not be correctly executed by the program interpreter. We are delighted to find out that our method reduces the error rate from $5.84\%$ to $3.82\%$, which indicates our method can predict more executable plan routines compared to the baseline VISPROG.

Table 3: Results on RefCOCO and RefCOCO+.

| Methods | IoU |
|---|---|
| Qwen-vl-chat-7b (Bai et al., 2023) | 32.54 |
| VISPROG (Gupta & Kembhavi, 2023) | 27.28 |
| EXOVIP | 31.50 |

Table 4: Visual reasoning on NLVR.

| Methods | Accuracy |
|---|---|
| OFA-large (Wang et al., 2022) | 58.38 |
| VISPROG (Gupta & Kembhavi, 2023) | 67.66 |
| EXOVIP | 67.96 |

Table 5: Abstract reasoning on KILOGRAM.

| Methods | Accuracy |
|---|---|
| CLIP-large (Radford et al., 2021) | 27.26 |
| VISPROG (Gupta & Kembhavi, 2023) | 24.46 |
| EXOVIP | 26.22 |

Table 6: Image editing on MagicBrush.

| Methods | CLIP-I | DINO |
|---|---|---|
| InstructPix2Pix (Brooks et al., 2022) | 84.19 | 69.60 |
| VISPROG (Gupta & Kembhavi, 2023) | 90.82 | 82.70 |
| EXOVIP | 91.27 | 83.40 |

### 5.2.2 VISUAL LANGUAGE GROUNDING

**Baseline Model**   We adopt the Qwen-vl-chat-7b (Bai et al., 2023) as the baseline. Qwen-vl-chat-7b is a pre-trained large vision-language model that uses Qwen-7B with further training with aligned techniques. Qwen-VL outperforms current SOTA generalist models on multiple VL tasks and has a more comprehensive coverage in terms of capability range.

**Analysis**   As demonstrated in Table 3, although our method can't achieve SOTA (Qwen-VL) on the RefCOCO dataset, it helps bridge the gap between VISPROG and the large vison-language model. While Qwen-VL is built on a LLM with 7 billion parameters, which is trained on trillions of tokens from the corpus, our method assembles a team of experts whose collective parameters total less than 1 billion. We believe our method can be improved with more advanced experts.

### 5.2.3 NATURAL LANGUAGE VISUAL REASONING

**Baseline Model**   We take the OFA-large (Wang et al., 2022) as baseline. OFA unifies a diverse set of cross-modal and unimodal tasks in a simple sequence-to-sequence learning framework.

**Analysis**   Table 4 shows the results. Although VISPROG exhibits strong complex reasoning ability over the end-to-end model, our method can hardly further improve its performance. We believe this is because we only take VQA modules to solve NLVR problems. The performance of decomposed VQA steps is hindered by the performance of VQA model, especially when there is error accumulation among a sequence of VQA steps.

### 5.2.4 VISUAL ABSTRACT REASONING

**Baseline Model**   We use the CLIP-large (Radford et al., 2021) as a baseline to test its performance on the text-to-image retrieval task proposed by KILOGRAM.

**Analysis**   For our method, given an object, we adopt the LLM to get its possible semantic parts. At the same time, we segment the image into several visual parts. After that, we align the semantic parts with the visual parts to enhance the matching process. In Table 5, we see the gap between VISPROG and CLIP. Although our method decreases the performance gap, the compositional method still can not achieve SOTA. Since part identification has already been demonstrated to play an important role in human abstraction Tversky & Hemenway (1984). We believe our method can be enhanced by introducing a better scene segmentation model.

### 5.2.5 TEXT-GUIDED IMAGE EDITING

**Baseline Model**   We take InstructPix2Pix (Brooks et al., 2022) as a baseline. InstructPix2Pix is a conditional diffusion model trained on GPT3 augmented datasets.

**Analysis**   Table 6 and Fig. 4 show the results on MagicBrush. These results illustrate the capability of our method to enhance the similarity between the edited image and the target image, signifying the precision of our image editing technique. For a more comprehensive evaluation of the editing quality, we have conducted a case study. Fig. 4 exhibits some instances using MagicBrush. While non-compositional methods are likely to change unrelated pixels, compositional methods are more con-

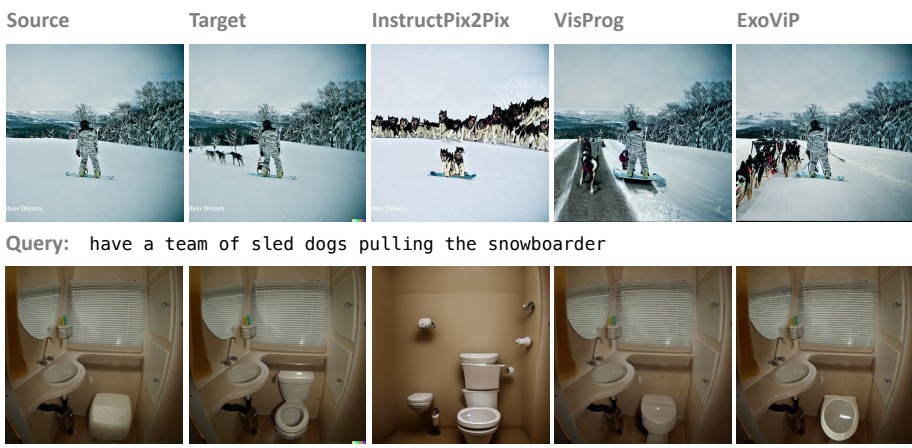

Figure 4: Qualitative results of text-guided image editing on MagicBrush.

trollable. Furthermore, when compared to VISPROG, our method excels in two key areas: accurately pinpointing the region that requires editing, and adjusting the image to the appropriate extent. This demonstrates the superiority of our method in both localization and modification of the image.

Table 7: Results for Open LLM on GQA.

| Methods | Accuracy |
|---------|----------|
| VISPROG (Llama-2-13b-chat) | 46.41 |
| EXOVIP ((Llama-2-13b-chat)) | 54.45 |

Table 8: Results for ViperGPT on GQA.

| Methods | Accuracy |
|---------|----------|
| ViperGPT (Dídac et al., 2023) | 45.47 |
| ViperGPT+ExoViP | 46.84 |

### 5.3 RESULTS ON OPEN LLMS

In this section, we present the results of applying our method to the open LLM. Specifically, we substituted GPT-3.5-turbo with LLama2-chat-13b (Touvron et al., 2023). The outcome of this substitution is displayed in Tab. 7. We are thrilled to discover that our method can yield significant improvements in open LLM.

### 5.4 GENERALIZABILITY OF OUR METHOD

To demonstrate the generalizability of our method, we apply our method to another compositional method, ViperGPT, which composes available modules by generating Python codes. We equip ViperGPT with our method and test its performance on the GQA dataset. We show the results in Table 8. We find the performance boost is less significant than which on VISPROG. We analyze this due to ViperGPT provides a few examples in the demonstration and it turns the parameter of the code-generation model to make it deterministic to generate subroutines. In other words, ViperGPT benefits little from our reasoning trace-searching strategy.

## 6 CONCLUSION

In this work, we identify two key types of errors in existing compositional methods: planning errors and module errors. To address these errors, we introduce an innovative verification framework EXOVIP. This framework verifies the correctness of vision module predictions. It corrects module errors by calibration and refines the planning process through tree searching. During this process, it considers both verification scores and the self-correctness of LLM. Applying the EXOVIP to two existing models, we achieve significant performance improvements across five different tasks. The results reinforce the promise and potential of EXOVIP on various open-world compositional reasoning tasks, marking an important milestone in the realm of multimodal tasks involving complex reasoning.

## ETHICS STATEMENT

The datasets referenced and utilized within our work are all publicly accessible, ensuring full transparency in our research process. We are dedicated to maintaining the highest ethical standards in all our undertakings, and we have ensured this by strictly adhering to the terms and conditions stipulated by the original licenses of these datasets.

## REPRODUCIBILITY STATEMENT

In this work, we provide the details of implementation in Sec. E. In addition, we provide the anonymous link [2], which includes a demo of our framework on the GQA dataset. Since we use the OpenAI API, *i.e.* gpt-3.5-turbo, people who would like to reimplement our work should get an API key first.

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
