# Appendices

## Table of Contents

## A  EXOSKELETON ALGORITHM

We demonstrate the overall algorithm of our method in Algorithm 1. There are mainly two parts: step-by-step verification and exploration with reasoning trace. To be more specific, we fuse the self-correctness ability of LLM into the procedure of tree-based reasoning trace searching, which has shown potential in calibrating the effectiveness of the searching algorithm.

## B  PROOF-OF-CONCEPT PILOT EXPERIMENTS

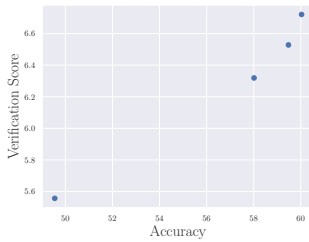

Figure 5: Accuracy on GQA positively correlates with the verification scores.

To evaluate the effectiveness of the verification modules, we try to find the relationship between verification scores and accuracy. All experiments are applied to the GQA dataset. We first disturb the examples in the demonstrations to get different plan results and corresponding verification scores. Specifically, we change the order of examples and select different portions of examples with four settings. After evaluation, we calculate the mean of verification scores of all steps. As is shown in figure 5, we are delighted to find the verification scores positively contribute to final accuracy. However, the trend is decreasing, which means when the verification scores increase to a certain extent, higher verification scores do little contribution to the final accuracy.

---

**Algorithm 1:** Exoskeleton Algorithm

---

**Input:** start step ($e_0$), goal node ($g$), scaling factor ($\tau$), verification size ($K$), rank size ($P$)
**Output:** Verified reasoning trace and intermedia results
$openList \leftarrow e_0$
$closedList \leftarrow empty\ list$
$path \leftarrow empty\ list$
**while** *open list is not empty* **do**
    $sort(openList, key = e_s)$
    Select top $K$ steps from $openList$ and put it in $closedList$ and empty $openList$
    $rank(closedList, key = LLM(e))$
    Select top $P$ steps to update $closedList$
    **for** $e\ in\ closedList$ **do**
        **if** $e\ is\ g$ **then**
            $path.add(e)$
            return $path$
        **else**
            $openList.add(e.next)$
        **end**
    **end**
    **for** $e\ in\ openList$ **do**
        $e_s = avg(e_s^{item} - e_n^{item}, e_s^{cap} - e_n^{cap}, e_s^{vqa} - e_n^{vqa})$
        $e \leftarrow Verify(NORM(e_s, \tau), e)$
    **end**
**end**

---

## C  ERROR ANALYSIS OF VISPROG AND EXOVIP

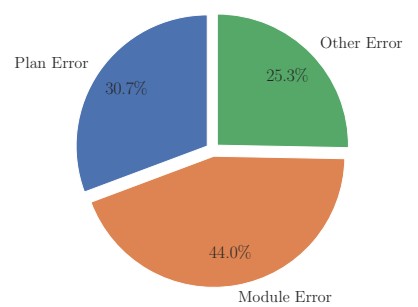

Figure 6: Distribution of the failure cases of original VISPROG.

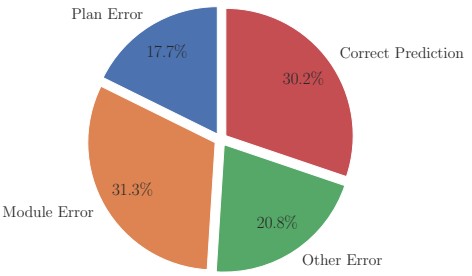

Figure 7: Distribution of the failure cases of EXOVIP.

We manually analyze 100 randomly sampled failure cases on VISPROG. We find that there are three typical reasons for the failures: (a) vision module prediction error; (b) LLM planning error; (c) others. We demonstrate the statistics of the failure cases in Fig. 6. Following the application of our proposed framework, we reassessed the same cases in Fig. 7 and were pleased to discover a reduction in module errors by $28.87\%$, and a decrease in planning errors by $42.35\%$. Nevertheless, our current strategy was unable to rectify $69.8\%$ of the errors. When juxtaposed with the data from Tab. 1, our method has enhanced VISPROG by $7.11\%$, which is lower than the improvement of the failure cases. This outcome suggests that our approach may give rise to novel challenges. We further demonstrate common errors of our method in Fig. 10 and Fig. 11. We find the majority of these failure cases originate from the VQA module.

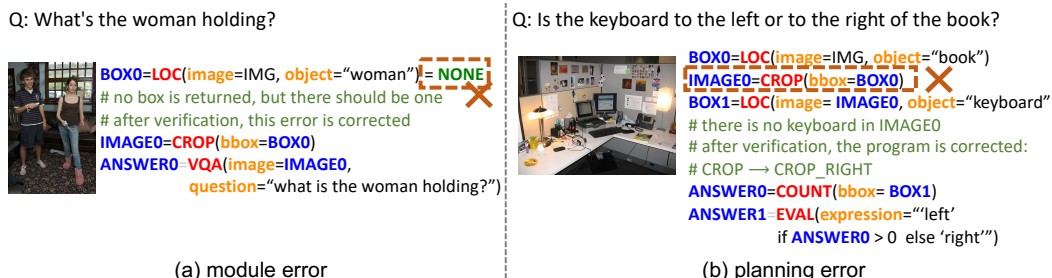

Figure 8: Existing methods suffer from two types of errors: (a) vision module prediction error and (b) LLM planning error. Our verification modules help correct the errors.

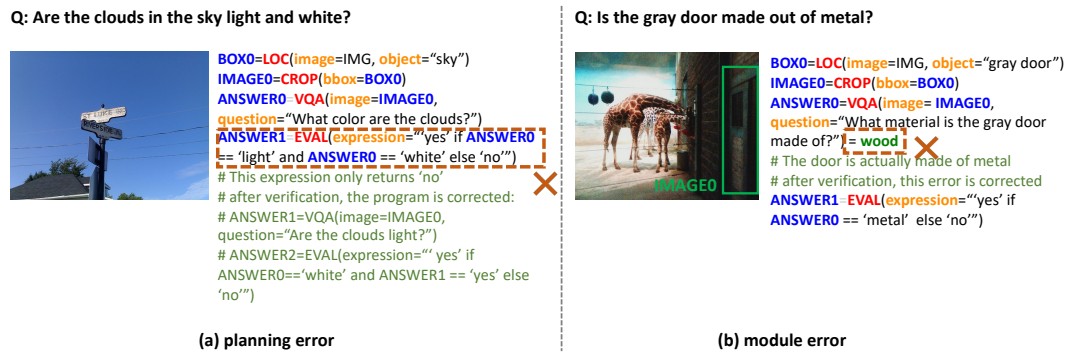

Figure 9: More examples of the two types of errors: (a) vision module prediction error and (b) LLM planning error.

In Figs. 8 and 9, we show examples of failure cases of the original VISPROG.

**Q: Are both the shoe and the cloud the same color?**

BOX0=LOC(image=IMAGE, object="shoe")
IMAGE0=CROP(image=IMAGE, bbox=BOX0)
BOX1=LOC(image= IMAGE, object="cloud")
IMAGE1=CROP(image= IMAGE, bbox=BOX1)
ANSWER0=VQA(image=IMAGE0, question="What color is the shoe?") ✗
# the result is wrongly predicted as white, current VQA model make a lot of errors on color recognition tasks.
ANSWER1=VQA(image=IMAGE1, question="What color is the cloud?")
ANSWER1=EVAL(expression="'yes' if ANSWER0 == ANSWER1 else 'no'") = yes

Figure 10: Common failure cases: some modules perform badly on certain tasks, *e.g.* the VQA module performs poorly on color recognition tasks.

**Q: Which place is it ?**

ANSWER0=VQA(image=IMAGE0, question="Which place is it?") = zoo  ✕
# The reference answer is forest

Figure 11: Common failure cases: some queries can not be decomposed into sub-tasks. Our method helps little with these non-decomposable queries.

# D  EFFICIENCY ANALYSIS

We present the average inference time on the GQA dataset. Generally, the temporal expenditure of our tree-based search method significantly surpasses that of VisProg. However, the majority of the time is consumed by the call of the OPENAI API, an issue we posit is intrinsic to analogous works Yao et al. (2023); Feng et al. (2023); Zhou et al. (2023). When compared to Depth First Search/ Breadth First Search Yao et al. (2023) or Monte Carlo Tree Search Feng et al. (2023); Zhou et al. (2023), we assert that our beam search-based method can achieve an optimal equilibrium between efficiency and effectiveness.

Table 9: Average Inference Time on the GQA Dataset

| Methods | Total Inference Time (s) | Planning Time (s) | Module Inference Time (s) |
|---|---|---|---|
| VISPROG | 1.59 | 1.10 | 0.49 |
| EXOVIP | 4.32 | 3.64 | 0.68 |

# E IMPLEMENTATION DETAILS

## E.1 VISUAL MODULES.

| Task | Operation Modules | | | | | Verification Modules | |
|---|---|---|---|---|---|---|---|
| Compositional Image QA | **LOC** OWL-ViT | **VQA** BLIP | **FILTER** CLIP | **COUNT** len() | **EVAL** eval() | **SIM** CLIP | **CAP** BLIP |
| | **CROP** PIL.crop() | **CROPLEFT** PIL.crop() | **CROPRIGHT** PIL.crop() | **CROPABOVE** PIL.crop() | **CROPBELOW** PIL.crop() | **VQA** BLIP | |
| Visual Grounding | **LOC** OWL-ViT | **FILTER** CLIP | **TAG** PIL.rectangle() | | | **SIM** CLIP | |
| Natural Language for Visual Reasoning | **VQA** BLIP | **EVAL** eval() | | | | **CAP** BLIP | **VQA** BLIP |
| Abstract Reasoning | **PART** ChatGPT | **SEG** Maskformer | **ALIGN** CLIP | **SELECT** CLIP | | **SIM** CLIP | |
| Text-guided Image Editing | **SEG** Maskformer | **SELECT** CLIP | **REPLACE** Stable Diffusion | | | **SIM** CLIP | |

Figure 12: The neural modules (green) and symbolic modules (pink) used in our experiments.

We summarize the operation modules and the verification modules of different tasks in Figure 12. In practice, the candidate neural modules include OWL-ViT (Minderer et al., 2022), CLIP (Radford et al., 2021), BLIP (Li et al., 2022b), ChatGPT, MaskFormer (Cheng et al., 2021), Stable Diffusion (Rombach et al., 2022). In order to validate the effectiveness of our method and eliminate the benefits of external knowledge such as more advanced vision-language models which are trained on larger datasets. Both operation modules and verification modules are selected from the same candidate neural module sets. In other words, not all modules are verified on the mixture of all three types of modules.

## E.2 LLM PROMPTS

We demonstrate the prompt for self-correctness of all five tasks.

```
You are a ranker for a planner who use the candidate modules to answer a question:
QUESTION, select the best solutions for answering the question
candidate modules include: LOC: detection, VQA: visual question answering, EVAL: use
logic operation, RESULT: wrap up the final result,
CROP/CROP_LEFTOF/CROP_RIGHTOF/CROP_FRONTOF/CROP_INFRONT/CROP_INFRONTOF/CROP_BEHIND/CROP_
AHEAD/CROP_BELOW/CROP_ABOVE: crop the image.
Current solutions:
0 PLAN
1 PLAN
If the modules in the solutions have better cause-and-effect relations, and more likely
to answer the question, please rank it first. If you are unsure, please keep the
original rank. Return sequence number of currently best solution, for example 0,1,2,3,
DO NOT RETURN ANYTHING ELSE EXCEPT FOR NUMBERS SPLIT by ,
```

Figure 13: Self-correctness prompt of compositional question answering.

```
You are a ranker for a planner who use the candidate modules to carry out the
instruction: QUESTION, select the best solutions for carrying out the instruction
candidate modules include: LOC: detection, FILTER: filter unrelated objects, TAG: tag
the object, RESULT: wrap up the final result
Current solutions:
0 PLAN
1 PLAN
If the modules in the solutions have better cause-and-effect relations, and more likely
to answer the question, please rank it first. If you are unsure, please keep the
original rank. Return sequence number of currently best solution, for example 0,1,2,3,
DO NOT RETURN ANYTHING ELSE EXCEPT FOR NUMBERS SPLIT by ,
```

Figure 14: Self-correctness prompt of visual grounding.

```
You are a ranker for a planner who use the candidate modules to evaluate the statement:
QUESTION, select the best solutions for evaluating the statement
candidate modules include: VQA: visual question answering, EVAL: use logic operation,
RESULT: wrap up the final result \n Current solutions
Current solutions:
0 PLAN
1 PLAN
If the modules in the solutions have better cause-and-effect relations, and more likely
to answer the question, please rank it first. If you are unsure, please keep the
original rank. Return sequence number of currently best solution, for example 0,1,2,3,
DO NOT RETURN ANYTHING ELSE EXCEPT FOR NUMBERS SPLIT by ,
```

Figure 15: Self-correctness prompt of natural language for visual reasoning.

```
You are a ranker for a planner who use the candidate modules to carry out the
instruction: QUESTION, select the best solutions for carrying out the instruction
candidate modules include: SEG: segmentation, SELECT: select most related object,
REPALCE: edit image, RESULT: wrap up the final result
Current solutions:
0 PLAN
1 PLAN
If the modules in the solutions have better cause-and-effect relations, and more likely
to answer the question, please rank it first. If you are unsure, please keep the
original rank. Return sequence number of currently best solution, for example 0,1,2,3,
DO NOT RETURN ANYTHING ELSE EXCEPT FOR NUMBERS SPLIT by ,
```

Figure 16: Self-correctness prompt of text-guided image editing.

```
You are a ranker for a planner who use the candidate modules to carry out the
instruction: QUESTION, select the best solutions for carrying out the instruction
candidate modules include: PART: take apart an object, SEG: segment, ALIGN: align object
with query, RESULT: wrap up the final result
Current solutions:
0 PLAN
1 PLAN
If the modules in the solutions have better cause-and-effect relations, and more likely
to answer the question, please rank it first. If you are unsure, please keep the
original rank. Return sequence number of currently best solution, for example 0,1,2,3,
DO NOT RETURN ANYTHING ELSE EXCEPT FOR NUMBERS SPLIT by ,
```

Figure 17: Self-correctness prompt of visual abstract reasoning.

### E.3 Details of Visual Abstract Reasoning

In Fig. 18, we demonstrate our implementation of compositional methods on KILOGRAM dataset. Given an image, we segment it into several parts. At the same time, we adopt LLM to parse the query to several components. After that, we match the visual and textual components by their semantic similarity. Finally, we take the alignment score to retrieve the best matched image.

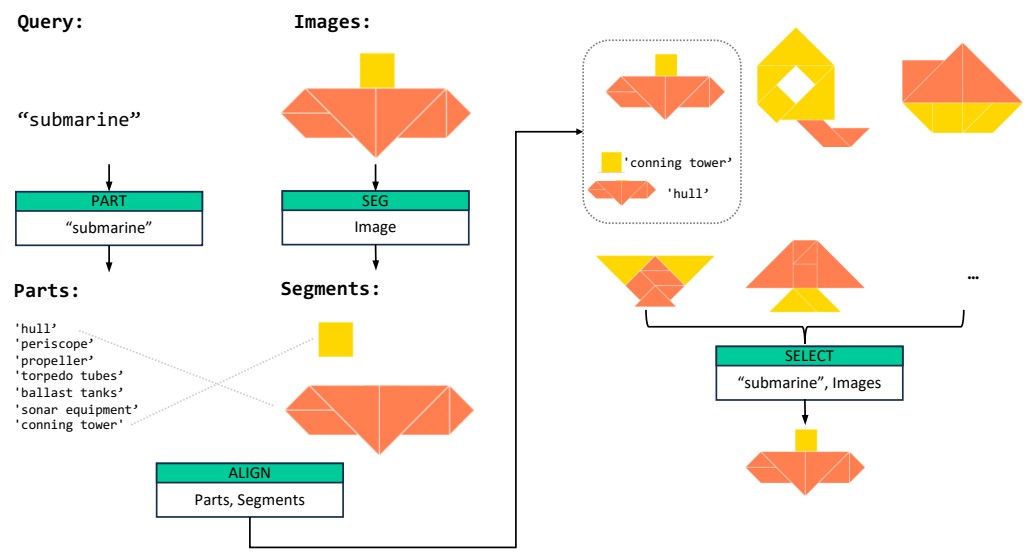

Figure 18: Implementation of abstract reasoning.

### E.4 Implementation details

In practice, for the verification modules, we set the $\tau$ as $2.0$ for $LOC$ module, $1.5$ for $SELECT$ module, and $ALIGN$ module, $1.2$ for other modules. For the negative sampling strategy, we select words sharing semantic similarity less than $0.5$ to construct the antonym vocabulary and randomly sample one antonym for each answer. In the searching process, we set up $K$ as $4$, and $P$ as $2$. To improve the efficiency of our search algorithm, we set the branching factor as $3$. To make the comparison fair, we use the same or fewer examples in the prompts for our methods, and select the verification modules from the operation modules. We apply our experiments on NVIDIA A100 GPU and NVIDIA 3090Ti GPU.

## F Qualitative study.

### F.1 Qualitative examples

We additionally exhibit more examples that can be improved by our method. As is shown in these examples, all five types of tasks could be further improved by our framework.

**Q: What material is the cup to the left of the laptop, plastic or glass?**

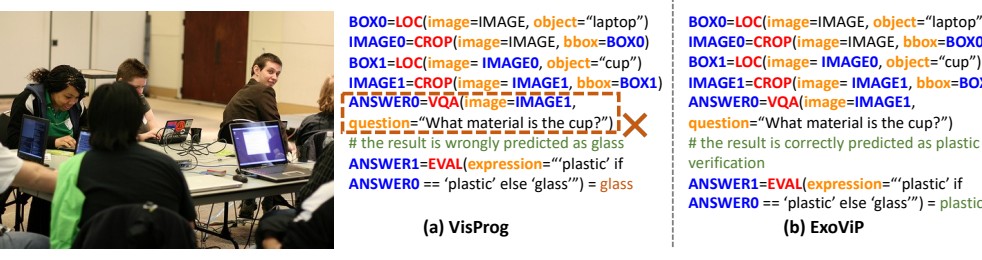

Figure 19: Qualitative study for GQA.

**Q: man in hat and robe**

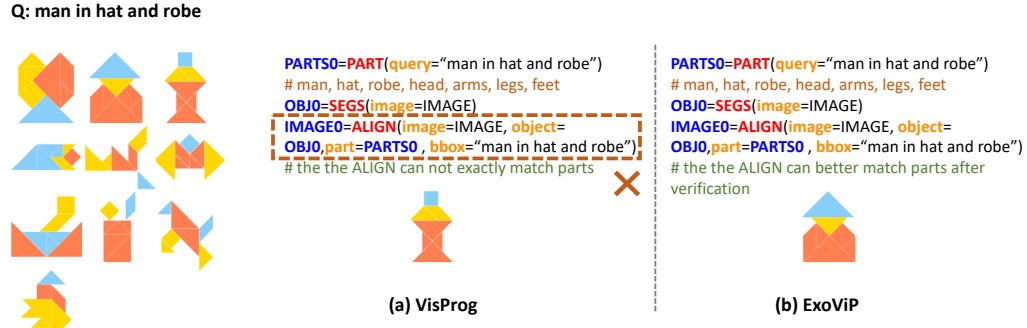

Figure 20: Qualitative study for KILOGRAM.

**Q: Tag the left zebra**

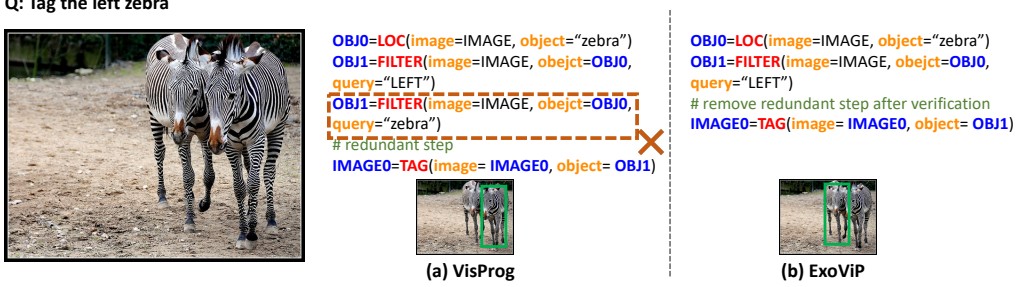

Figure 21: Qualitative study for RefCOCO.

**Q: The left image features a single fur-trimmed fingerless mitten with small embellishments dotting its front, and the right image shows a pair of fur-trimmed half-mitts with no thumb part showing.**

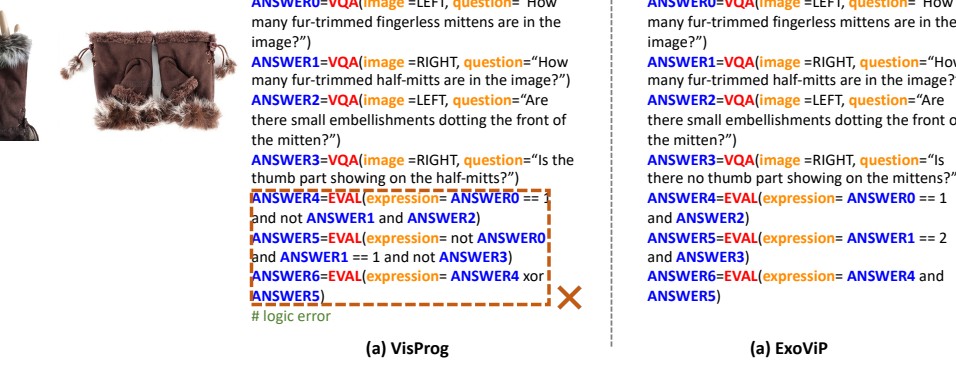

**ANSWER0**=**VQA**(**image** =LEFT, **question**="How many fur-trimmed fingerless mittens are in the image?")
**ANSWER1**=**VQA**(**image** =RIGHT, **question**="How many fur-trimmed half-mitts are in the image?")
**ANSWER2**=**VQA**(**image** =LEFT, **question**="Are there small embellishments dotting the front of the mitten?")
**ANSWER3**=**VQA**(**image** =RIGHT, **question**="Is the thumb part showing on the half-mitts?")
**ANSWER4**=**EVAL**(**expression**= **ANSWER0** == 1 and not **ANSWER1** and **ANSWER2**)
**ANSWER5**=**EVAL**(**expression**= not **ANSWER0** and **ANSWER1** == 1 and not **ANSWER3**)
**ANSWER6**=**EVAL**(**expression**= **ANSWER4** xor **ANSWER5**)
\# logic error ✗

**(a) VisProg**

**ANSWER0**=**VQA**(**image** =LEFT, **question**="How many fur-trimmed fingerless mittens are in the image?")
**ANSWER1**=**VQA**(**image** =RIGHT, **question**="How many fur-trimmed half-mitts are in the image?")
**ANSWER2**=**VQA**(**image** =LEFT, **question**="Are there small embellishments dotting the front of the mitten?")
**ANSWER3**=**VQA**(**image** =RIGHT, **question**="Is there no thumb part showing on the mittens?")
**ANSWER4**=**EVAL**(**expression**= **ANSWER0** == 1 and **ANSWER2**)
**ANSWER5**=**EVAL**(**expression**= **ANSWER1** == 2 and **ANSWER3**)
**ANSWER6**=**EVAL**(**expression**= **ANSWER4** and **ANSWER5**)

**(a) ExoViP**

Figure 22: Qualitative study for NLVR.