# OpenReview forum: "ExoViP: Step-by-step Verification and Exploration with Exoskeleton Modules for Compositional Visual Reasoning"
_ICLR.cc/2024/Conference — Submitted to ICLR 2024_

### Official Review · Reviewer_rVcU · 2023-10-30

**Soundness:** 4 excellent
**Presentation:** 3 good
**Contribution:** 4 excellent
**Rating:** 6
**Confidence:** 2

**Summary:**

Compositional visual reasoning methods, which strive to convert a complex query into a structured composition of manageable visual tasks, have demonstrated considerable potential in intricate multimodal tasks such as visual question answering and language-guided image editing. This paper introduces EXOVIP, a plug-and-play method designed to rectify errors at both the planning and execution stages. Empirical results from two representative vision-language programming methods exhibit consistent enhancements across five compositional reasoning tasks on a standard benchmark.

**Strengths:**

1. EXOVIP designs a mixture of three sub-verifiers, including an image-text matching verifier, an image captioning verifier, and a visual
question answering verifier, which seems to be reasonable.
2. A reasoning trace tree based on the verification scores as well as the self-correctness score from LLMs is also built to find the best trace that has the highest score.
3. To demonstrate the effectiveness of EXOVIP, this paper conducts experiments on two recent visual programming methods: self-defined programs, i.e., VISPROG and Python code programs, i.e., ViperGPT.

**Weaknesses:**

1. Detailed information about different prompts is necessary for a comprehensive comparison

2. The improvements on the GQA dataset appear to be minor when compared to those achieved by ViperGPT

**Questions:**

See weakness

---

> ### Author Response · Authors · 2023-11-14
> **Response to Reviewer rVcU**
>
> Thank you for your meticulous review and acknowledgment of our method's effectiveness. We believe your support of our work will help us to make a timely contribution and broader impact on both Computer Vision and compositional reasoning community. We would like to provide further clarification to address your concerns.
>
> > W1: Detailed information about different prompts is necessary for a comprehensive comparison
>
> **A:** In our submission, we have demonstrated the exact prompt for self-correctness for all five tasks, Please refer to **Appendix G** for details.
> Specifically, for example, for compositional question answering, the prompt goes like:
> `You are a ranker for a planner who use the candidate modules to answer a question: QUESTION, select the best solutions for answering the question \n candidate modules include: LOC: detection, VQA: visual question answering, EVAL: use logic operation, RESULT: wrap up the final result, CROP/CROP_LEFTOF/CROP_RIGHTOF/CROP_FRONTOF/CROP_INFRONT/CROP_INFRONTOF/CROP_BEHIND/CROP_AHEAD/CROP_BELOW/CROP_ABOVE: crop the image \n Current solutions: \n 0 PLAN \n 1 PLAN \n If the modules in the solutions have better cause-and-effect relations, and more likely to answer the question, please rank it first. If you are unsure, please keep the original rank. Return sequence number of currently best solution, for example 0,1,2,3, DO NOT RETURN ANYTHING ELSE EXCEPT FOR NUMBERS SPLIT by ,`
>
> We are willing to append more details if there is any further missed information.
>
>
> > W2: The improvements on the GQA dataset appear to be minor when compared to those achieved by ViperGPT
>
> **A:** Thank you for bringing this to our attention. We would like to highlight 4 aspects:
>
> - Our test set differs from that of ViperGPT. We used the same sampling strategy as in VisProg. As is claimed in **Section 5.1**, `we randomly sample 5 samples from the balanced val set and 20 samples from the testdev set of each question type. e.g. “weatherVerify” for judging the weather, “twoCmomon” for judging common attributions of two objects. In summary, there are 102 question types and 2327 questions in our test set.`
> - Regarding the different test sets, we re-executed ViperGPT's experimental results on our test set using their official repository (https://github.com/cvlab-columbia/viper) without any modifications. The results indicate that VisProg+ExoViP (61.49%) improves the performance of ViperGPT (46.47%) by 5.02%.
> - Our insights differ from ViperGPT. While ViperGPT has transitioned to Python code for planning, thereby inherently addressing numerous planning errors, our module is designed to enhance existing compositional methods by tackling both planning and module errors.
> - As shown in Table 7, ExoViP can be generalized to improve ViperGPT by 1.37%.

---

> > ### Author Response · Authors · 2023-11-15
> > **Looking forward to your feedback**
> >
> > Dear rVcU,
> >
> > We appreciate your positive score and valuable suggestions on our work. We also appreciate that you and other reviewers all noticed the effectiveness of our work. As acknowledged by Reviewer Tiuv, 2ndj, atdm, our method is `innovative`, `well motivated`, and `marks a notable advancement in the field` (Refer to https://openreview.net/forum?id=8Xx0mKoCMd&noteId=efvjr2Jzgd for the summary of our contributions).
> >
> > We've tried our best to provide clarifications to your questions. Please kindly let us know if you have further comments. We will be more than happy to answer.
> >
> > Best,
> >
> > Authors

---

### Official Review · Reviewer_atdm · 2023-11-10

**Soundness:** 2 fair
**Presentation:** 2 fair
**Contribution:** 2 fair
**Rating:** 5
**Confidence:** 3

**Summary:**

This paper points out two major challenges when using Visual Programming, namely 1) module error and 2) planning error. The authors propose a plug-and-plan method, EXOVIP, which can self-correct the errors on the planning side and on the module execution side. More particularly, three verification module (e..g, Image-text matching, Image captioning, and VQA verifier) are proposed to verify each step. In the experiments, they show improvement of EXOVIP over BLIP2/InstructBLIP/VISPROG on GQA, and over Qwen-v1-chat-7b/VISPROG on RefCOCO, InstructPix2Pix/VISPROG on MagicBrush, etc.

**Strengths:**

1. The main idea is clear, and it's worth diving into this area to mitigate planning error and module error. The proposed verification score for prediction calibration is straightforward and show some improvements on several datasets.
2. The paper are written clearly in general, though some small typos. The figures clearly present the intuition behind the proposed method.

**Weaknesses:**

1. The verifier module is not carefully developed, CLIP for image-text matching is particularly not good at give correct similarity score, especially when the task are for compositional visual reasoning in this paper. In addition BLIP version1 (as cited in Section 4.1) for image caption/VQA is also not SOTA, and no ablation study is explored here to choose the best module for verifiers.
2. The introduced verifiers can cost the whole system more computation time, however no efficiency info is added here, it should be more fair to consider this. Besides, the sub-verifiers for calibration introduce propagation error which is not handled, and it's not clear when the verification score is more reliable than the confidence score, while in the method, the verification score directly replace it.
3. Very concerned with the compared baselines. e.g., many recent MLLMs achieve better results on GQA than the baselines used in Table 1, e.g., LLaVA get 63 on it. In Table 3, why is the reported referring expression comprehension results only on RefCOCO and RefCOCO+, why not including RefCOCOg, which is quite a standard comparison for this task. And why only a subset is validated, 2 samples per type from the test set is actually not enough. And the compared model and proposed model are only 30 IoU, which is far below the decent results in recent MLLM, which all receive 80+ (e.g., Shikra, Kosmos2, etc)
4. In conclusion, the main concern is the baselines compared in each Table, very confusing, a lot SOTA models are removed out e.g., OFA-large is selected for NLVR, but not included for RefCOCO (which is actually reported in the origin paper). And the proposed method is not carefully developed or verified, as not sufficient ablation for each module is conducted, especially the choice for each verifier.

**Questions:**

1. For the three proposed verifiers, the CLIP is used for image-text matching, and BLIP is used for image captioning and VQA, how these models are selected, is there any ablation experiments here? E.g., C_img is generated by BLIP, why not using SOTA caption model, at least BLIPv2? Is this a severe bottleneck for the verifier or even bring confusion when the verifier model make errors.
2. It happens when the confidence score is more accurate than the verification score. However, seems in candidate prediction in Equation 6, is only using the averaged verification score to calibrate, while the confidence score is ignored.
3. How is the propagation error being controlled? The verifier is proposed to mitigate the planning error and module error. How is the newly. introduced verifier module error being handled?

---

> ### Author Response · Authors · 2023-11-14
> **Response to Reviewer atdm (1/3)**
>
> Thank you for your careful review and insightful comments.  We have meticulously discussed the issues at hand and have attempted to dispel any misconceptions through our detailed responses. We trust that our comprehensive, point-by-point rebuttals have satisfactorily addressed all your concerns. Please let us know if there are further questions and we are delighted to have a deeper discussion.
>
>
> >W1: The verifier module is not carefully developed, CLIP for image-text matching is particularly not good at give correct similarity score, especially when the task are for compositional visual reasoning in this paper. In addition BLIP version1 (as cited in Section 4.1) for image caption/VQA is also not SOTA, and no ablation study is explored here to choose the best module for verifiers.
>
> >W3: And the proposed method is not carefully developed or verified, as not sufficient ablation for each module is conducted, especially the choice for each verifier.
>
> >Q1: For the three proposed verifiers, the CLIP is used for image-text matching, and BLIP is used for image captioning and VQA, how these models are selected, is there any ablation experiments here? E.g., C_img is generated by BLIP, why not using SOTA caption model, at least BLIPv2? Is this a severe bottleneck for the verifier or even bring confusion when the verifier model make errors.
>
>
> **A:** Sorry for any confusion. We have carefully considered the selection of models for each verifier module. Specifically,
> 1. **Fair module comparisons.** We intentionally selected modules from VisProg's original modules. As noted in the implementation detail section of **Appendix E**, "` In order to validate the effectiveness of our method and eliminate the benefits of external knowledge such as more advanced vision-language models which are trained on larger datasets. Both operation modules and verification modules are selected from the same candidate neural module sets.`".
> 2. **Trade-off between latency and performance.** As claimed in **Section 5.2.2**, "`while Qwen-VL is built on a LLM with 7 billion parameters, which is trained on trillions of tokens from the corpus, our method assembles a team of experts whose collective parameters total less than 1 billion`". Our goal is not only to achieve better performance with heavier and more models, but to deliver a carefully designed system that is both lightweight and easy to plug-and-play, offering superior performance with minimal additional computational expense. Specifically, when we substitute the QA model (blip-vqa-capfilt-large) with BLIPv2 (blip2-flan-t5-xl), we see a performance increase from 67.96 to 68.70 on NLVR. However, this comes with a *tenfold increase in parameters*.
>
>
> > W2: The introduced verifiers can cost the whole system more computation time, however no efficiency info is added here, it should be more fair to consider this.
>
> **A:** Thank you for pointing it out. We present the mean inference time on the GQA dataset. Generally, the temporal expenditure of our tree-based search method significantly surpasses that of VisProg. However, the majority of the time is consumed by the call of the OPENAI API, an issue we posit is intrinsic to analogous works [1,2,3]. When compared to Depth First Search/ Breadth First Search [1] or Monte Carlo Tree Search [2,3], we assert that our beam search-based method can achieve an optimal equilibrium between efficiency and effectiveness. We have added this part to **Appendix D** in our revised paper.
>
> | model   | Total Inference Time (s) |  Planning Time (s)   | Module Inference Time (s) |
> | ------- | -------------------- | --- | -------------------------- |
> | VisProg | 1.59                 |  1.10   | 0.49                       |
> | ExoViP  | 4.32                 |   3.64  | 0.68                       |
>
>
>
>
> >W2: It's not clear when the verification score is more reliable than the confidence score, while in the method, the verification score directly replace it.
>
> >Q2: It happens when the confidence score is more accurate than the verification score. However, seems in candidate prediction in Equation 6, is only using the averaged verification score to calibrate, while the confidence score is ignored.
>
> **A:** Sorry about the confusion. There might be severe misunderstandings regarding our method. we never used the verification score to replace the confidence score. In Equation 6, $p$ represents the confidence score, while $w$ denotes the normalized verification score, also known as the weight. Intuitively, the verification score, or weight, is used to scale (calibrate) the confidence score. We have not ignored the confidence score at any point.

---

> > ### Author Response · Authors · 2023-11-14
> > **Response to Reviewer atdm (2/3)**
> >
> > >W2: Besides, the sub-verifiers for calibration introduce propagation error which is not handled.
> >
> > >Q3: How is the propagation error being controlled? The verifier is proposed to mitigate the planning error and module error. How is the newly. introduced verifier module error being handled?
> >
> > **A:** Apologies for any confusion. There are two key components in our method specifically designed to mitigate the problem.
> >
> > - **Negative Sampling Strategy**: We calculate the verification score as the difference between candidates and their negative samples. This score serves as a verification score and is used to calibrate the confidence score. As outlined in **Section 4.1**, we discovered that without this strategy,` directly applying this verification score does not work well.` Therefore, `Motivated by recent works in truthfulness [4], commonsense [5], and bias [6], ` We proposed the negative sampling strategy. Our method is also partially inspired by recent developments in multimodal counterfactuality [7, 8]. The experimental results shown in Table 1 further demonstrate the effectiveness of this negative sampling strategy. In other words, we adopt negative sampling strategy to prevent propagation errors introduced by verifiers.
> > - **Combination of Verification Score and Self-correctness of LLMs**: To improve planning results as well as the robustness of our method, we incorporate the verification score with self-correctness of LLMs in the tree-based search algorithm. As claimed in **Section 4.2**, `because the verification scores can be very close for the selected K traces, we further use the self-correctness ability of LLMs to reorder the K traces and select the top P from them (P < K).` The experimental results shown in Table 1 further demonstrate the effectiveness of the combination. In other words, we leverage the self-correctness of LLMs to prevent propagation errors introduced by verifiers.
> >
> >
> >
> > > W3: Very concerned with the compared baselines.
> >
> > **A:** We would like to clarify the selection of both a subset of datasets and baseline models.
> > - **Dataset:** As we mentioned in **Section 5.1**, "`becasue VISPROG doesn’t release their sampled evaluation subset, we do sampling following the VISPROG paper and evaluate all the methods on our sampled evaluation set.`" For a subset of the datasets, we follow the same rationale as VISPROG. "`In order to limit the money spent on generating programs with GPT-3, we create a subset of GQA
> > for evaluation.`" We also align with VISPROG in terms of the number of datasets we use. Specifically, for the Factual Knowledge Object Tagging task, they have "`100 tagging instructions`", and for the Image Editing task, they have "`107 instructions`".
> > - **Baselines:** Our insights are different with MLLMs. While most of them are designed to incorporate multimodal instructions into LLMs to achieve SOTA on QA tasks in general domain (e.g. [MME](https://github.com/BradyFU/Awesome-Multimodal-Large-Language-Models/tree/Evaluation), [MMBench](https://opencompass.org.cn/leaderboard-multimodal), [SEED-Bench](https://huggingface.co/spaces/AILab-CVC/SEED-Bench_Leaderboard)),  our method is specifically designed to address the issues of existing visual compositional methods and to better integrate MLLMs into our framework. To clarify, in our experimental setting, we have not been able to find a single MLLM that is suitable for all five tasks. Consequently, we have to select parts of the SOTA models for reference. As we mentioned in **Section 1**, all the experiments are designed to "`show consistent improvements with the two models on the five tasks`". It's important to note that we have not claimed to achieve a state-of-the-art result in this work. In response to your request, we will endeavor to incorporate additional baseline results in the subsequent discussion.

---

> ### Author Response · Authors · 2023-11-14
> **Response to Reviewer atdm (3/3)**
>
> >W3: many recent MLLMs achieve better results on GQA than the baselines used in Table 1, e.g., LLaVA get 63 on it.
>
> **A:** Thank you for the references. We have included the results of LLava, as an upper bound of non-compositional models, in Table 1.
>
> As claimed in the introduction, "`despite their merits, current visual programming methods still suffer from challenges due to the failure of the LLM planning or the visual modules, lagging behind the performance
> of non-compositional models.`", the focus of our work is to mitigate the performance gap between compositional models and non-compositional ones (e.g. LLaVA). Therefore, models like LLaVA can only be considered upper-bound and lead to unfair comparisons to our model.
> Specifically, the result of 63.0 on GQA ("`e.g., LLaVA get 63 on it`") is from Llava (Llava-1.5-13B) [9]. However, this result is derived from instructions that were tuned on the GQA training corpus, making it incompatible with the zero-shot setting of our experiments. Additionally, the evaluation scripts for Llava come with an extra prompt, "\nAnswer the question using a single word or phrase.". This particular prompt is not included in the scripts for other baselines.
>
>
>
> | Model | Accuracy on GQA |
> | -------- | -------- |
> | Llava-1.5-13b | 74.56 |
> | VISPROT     |  57.41    |
> | ExoViP     |   61.49   |
>
>
>
> >W3: In Table 3, why is the reported referring expression comprehension results only on RefCOCO and RefCOCO+, why not including RefCOCOg, which is quite a standard comparison for this task.
>
> **A:** Thank you for your suggestion. Due to limited resources and funds, we are unable to apply our method to all existing benchmarks. However, in response to your request, we will endeavor to report the results on the subset RefCOCOg, which is sampled by the same strategy as RefCOCO and RefCOCO+.
>
> | Model | IoU on RefCOCOg |
> | -------- | -------- |
> | Qwen-vl-chat-7b     |  18.53   |
> | VISPROG     |   14.83   |
> | ExoViP     |   15.82  |
>
>
> >W3:  And why only a subset is validated, 2 samples per type from the test set is actually not enough.
>
> **A:** As previously mentioned, our financial constraints limit us from affording a complete test set. However, the scale of our dataset is comparable to that of VISPROG. We also believe that implementing a balanced sampling strategy is of utmost importance.
>
>
> >W3: And the compared model and proposed model are only 30 IoU, which is far below the decent results in recent MLLM, which all receive 80+ (e.g., Shikra, Kosmos2, etc).
>
>
> **A:** It is worth noting that the testing dataset used to report evaluation results of `Shikra, Kosmos, etc.` is different from the evaluation dataset used in our work (following the sampling strategy as in VisProg; refer to **Sec. 5.1 GQA setup** for details). In order to stay aligned with the current advancements in MLLM, we have included some preprint models in our experiment, such as QWen-vl-7b-chat. To the best of our knowledge, this model is a state-of-the-art MLLM during our experimental period, and it outperforms Shikra in terms of results on RefCOCO. We have used the model weight from its official code repository to evaluate our dataset (URL: https://github.com/QwenLM/Qwen-VL): as reported in our paper, it achieves a score of 32.54 in the subset of RefCOCO&RefCOCO+. We are willing to include more references and compare with them if there are more advanced models. We have also updated all the evaluation scripts in the anonymous link provided in the submitted paper.
>
>
> >W3: In conclusion, the main concern is the baselines compared in each Table, very confusing, a lot SOTA models are removed out e.g., OFA-large is selected for NLVR, but not included for RefCOCO (which is actually reported in the origin paper).
>
> **A:** As previously stated, our objective in this work is not to achieve the SOTA performance on all tasks. We have selectively referenced parts of the SOTA models. Since Qwen-vl-7b significantly outperforms OFA-Large on the referring task, we did not report its results.
>
>
>
>
>
>
> [1] Tree of Thoughts: Deliberate Problem Solving with Large Language Models
>
> [2] Alphazero-like Tree-Search can Guide Large Language Model Decoding and Training
>
> [3] Language Agent Tree Search Unifies Reasoning Acting and Planning in Language Models
>
> [4] Mitigating Lies in Vision-Language Models
>
> [5] Improving Commonsense in Vision-Language Models via Knowledge Graph Riddles
>
> [6] A Multi-dimensional study on Bias in Vision-Language models
>
> [7] Multimodal Explanations by Predicting Counterfactuality in Videos
>
> [8] Towards Counterfactual Image Manipulation via CLIP
>
> [9] Improved Baselines with Visual Instruction Tuning

---

> > ### Author Response · Authors · 2023-11-15
> > **Looking forward to your feedback**
> >
> > Dear Reviewer atdm,
> >
> > We thank you again for your valuable comments and suggestions to help us improve our paper. We appreciate that you acknowledged that `it's worth diving into this area to mitigate planning error and module error.`
> >
> > We've tried our best to address all your concerns **point-by-point** and made substantial modifications to our manuscripts. It would be more than helpful if you would take some time to read our responses and updated paper. Please feel free to let us know if you have further questions and we would be happy to respond.
> >
> > Best,
> >
> > Authors

---

### Official Review · Reviewer_2ndj · 2023-11-10

**Soundness:** 4 excellent
**Presentation:** 3 good
**Contribution:** 2 fair
**Rating:** 5
**Confidence:** 5

**Summary:**

The paper presents a novel approach called "exoskeleton" verification modules designed for visual programming. These modules methodically validate the accuracy of vision module predictions. They are used to rectify errors in module predictions through calibration, and in planning, using tree searching techniques. This involves incorporating verification scores and the self-correction capability of large language models (LLMs). The approach is tested on five different tasks, demonstrating its efficacy. Notably, when applied to the VISPROG model in a compositional visual question answering task, the method achieved a notable 4% increase in accuracy.

**Strengths:**

This paper introduces an innovative concept in the field of visual programming - the "exoskeleton" verification modules. This unique approach effectively tackles both module and planning errors in compositional visual reasoning.
The robustness of the proposed methodology is evident through its application to two distinct models, yielding a notable 4% improvement in accuracy for compositional visual question answering tasks. Furthermore, the paper conducts a thorough preliminary study, demonstrating how individual components (such as self-correctness, beam search, and verification) contribute to enhanced performance.
The paper excels in clearly articulating its methodology. It provides detailed explanations on the use of exoskeleton modules for verification processes and the precise methods employed in calibrating module predictions.
The contributions of this paper are highly significant for the field of visual compositional reasoning. It addresses critical limitations in existing methodologies and offers a substantial improvement in prediction accuracy, marking a notable advancement in the field.

**Weaknesses:**

The paper's experimental evaluation is somewhat limited, primarily focusing on VISPROG evaluation datasets. Expanding the range of datasets, especially including more varied and challenging ones, would significantly strengthen the demonstration of the method's generalizability. While the paper reports a 4% improvement on the GQA dataset, a broader dataset spectrum could provide a more comprehensive assessment of the method's effectiveness.
The paper lacks an in-depth analysis of its limitations and potential failure cases. For visual language grounding, natural language reasoning, and visual abstract reasoning tasks where the model fails to exceed the performance of SOTA models, no further analysis is provided apart from reporting the results. Although several failure cases are mentioned in the attachment, they are not thoroughly investigated or analyzed. A detailed examination of these cases would be invaluable for identifying areas for further refinement and development of the method.
The paper’s approach appears to primarily involve an ensemble of multiple vision-language (VL) models for post-correction, which could be perceived more as an engineering endeavor based on the VISPROG work rather than a novel conceptual contribution. A more distinct delineation of the method's innovative aspects compared to existing VL models would be beneficial in highlighting its unique contributions to the field.

**Questions:**

Generalizability Across Datasets:
Question: Could you elaborate on the choice of primarily using VISPROG evaluation datasets?
Suggestion: Consider applying your method to a wider array of datasets, especially those that are more diverse and challenging. This could help in better demonstrating the method's generalizability and effectiveness across different scenarios.

Analysis of Limitations and Failure Cases:
Question: Why was there no detailed analysis provided for the failure cases mentioned in the attachment?
Suggestion: It would be beneficial to include a thorough investigation and analysis of these failure cases. This analysis could offer insights into the method's limitations and guide future improvements.

Clarification on Novelty and Conceptual Contributions:
Question: How does the proposed method differentiate itself, in terms of novelty, from simply being an ensemble of multiple VL models for post-correction?
Suggestion: A more explicit explanation of the novel aspects of your approach would be helpful. Clarifying how your method advances beyond just combining existing VL models could strengthen the paper's claims of originality.

---

> ### Author Response · Authors · 2023-11-14
> **Response to Reviewer 2ndj (1/2)**
>
> Thank you for your insightful review and kind words regarding our innovative concept of verification modules. We appreciate your recognition of the **robustness of our method** and your acknowledgment of our **efforts to address critical limitations in existing methodologies**. We have carefully followed your suggestion to revise our paper. Please let us know if you have any more comments and we are willing to answer more details and further improve our work.
>
> >W: The paper's experimental evaluation is somewhat limited, primarily focusing on VISPROG evaluation datasets. Expanding the range of datasets, especially including more varied and challenging ones, would significantly strengthen the demonstration of the method's generalizability. While the paper reports a 4% improvement on the GQA dataset, a broader dataset spectrum could provide a more comprehensive assessment of the method's effectiveness.
>
> > Q1: Could you elaborate on the choice of primarily using VISPROG evaluation datasets? Suggestion: Consider applying your method to a wider array of datasets, especially those that are more diverse and challenging. This could help in better demonstrating the method's generalizability and effectiveness across different scenarios.
>
> **A:** Thank you for your kind advice. We choose similar tasks as in VisProg for fair comparison and better demonstration of the improvements. In addition, we have added an abstract reasoning task KILOGRAM, `a dataset featuring extensively annotated tangram
> puzzles that necessitate the model’s comprehension of abstract tangram shapes and their subsequent
> classification`. This is a challenging task ` designed
> to evaluate the model’s capacity to abstractly generalize, utilizing visually ambiguous stimuli to do
> so`. In **Section 5.2.4**, we demonstrate `how our method effectively utilizes conceptual components to enhance abstract image understanding.` Furthermore, in **Appendix E**, we present the details of our implementation of abstract reasoning.
>
>
> >W: The paper lacks an in-depth analysis of its limitations and potential failure cases. For visual language grounding, natural language reasoning, and visual abstract reasoning tasks where the model fails to exceed the performance of SOTA models, no further analysis is provided apart from reporting the results. Although several failure cases are mentioned in the attachment, they are not thoroughly investigated or analyzed. A detailed examination of these cases would be invaluable for identifying areas for further refinement and development of the method.
>
> >Q2: Why was there no detailed analysis provided for the failure cases mentioned in the attachment? Suggestion: It would be beneficial to include a thorough investigation and analysis of these failure cases. This analysis could offer insights into the method's limitations and guide future improvements.
>
> **A:** Thank you for your valuable advice. We have reassessed the failure cases in GQA. Both the statistical results and qualitative results are now presented in **Appendix C** of our revised paper. Additionally, supplementing the analysis in **Section 5.2**, we analyze the failure cases of tasks that fail to exceed the performance of SOTA models.
>
> In conclusion, We have the following observations:
> 1. Regarding the sampled failure cases in GQA, we observed a reduction in module errors by 28.87% and a decrease in planning errors by 42.35%. Our method seems to be more effective in resolving planning errors. We believe this improvement is due to the incorporation of verification and self-correction mechanisms in our tree-based search algorithm. In essence, there is a need to enhance the verification process for candidate predictions.
> 2. In the GQA task, our method has corrected 31.2% of the failure cases. However, when compared to the general improvements on the test set, we found that our method also introduced new issues. Therefore, we analyzed the failure cases specific to our method and discovered that it tends to introduce a planning trace made of the VQA modules, which often perform poorly. This observation has led us to consider introducing penalties for certain modules that perform poorly.
> 3. Upon analyzing failure cases in visual language grounding, natural language visual reasoning, and visual abstract reasoning tasks, where the model falls short of outperforming SOTA models, we identified two attributes in these reasoning traces that could contribute to subpar performance: (a) the traces for visual abstract reasoning and visual language grounding are relatively short, making few improvements by our verification modules, and (b) the traces for natural language visual reasoning are primarily composed of VQA modules, which could potentially serve as bottlenecks.

---

> > ### Author Response · Authors · 2023-11-14
> > **Response to Reviewer 2ndj (2/2)**
> >
> > >W: The paper’s approach appears to primarily involve an ensemble of multiple vision-language (VL) models for post-correction, which could be perceived more as an engineering endeavor based on the VISPROG work rather than a novel conceptual contribution. A more distinct delineation of the method's innovative aspects compared to existing VL models would be beneficial in highlighting its unique contributions to the field.
> >
> >
> > >Q3: How does the proposed method differentiate itself, in terms of novelty, from simply being an ensemble of multiple VL models for post-correction? Suggestion: A more explicit explanation of the novel aspects of your approach would be helpful. Clarifying how your method advances beyond just combining existing VL models could strengthen the paper's claims of originality.
> >
> > **A:** Thank you for the suggestion. We summarize major differences, in terms of novelty, between ExoViP and simply combining VL models for post-correction:
> > - **Mixture of VL Modules from Different Dimensions**: We deliberately select three modules for verification, each offering a unique dimension. In **Section 4.1**, we highlight their distinct characteristics. (i) Image-text matching verifier `returns the semantic representation of the image-sentence pair`; (ii) Image caption verifier `describes the visual details of the image`; (iii) Visual question-answering `evaluates the advanced relationships between image and language, such as entailment and factual consistency`.
> > - **Negative Sampling Strategy**: The verification score is calculated as the difference between candidates and their negative samples. This concept is partly inspired by recent developments in multimodal counterfactuality [1, 2]. Moreover, as outlined in **Section 4.1**, we discovered that without this strategy,` directly applying this verification score does not work well.`
> > - **Calibration Using Verification Scores**: The verification score is used to calibrate the results of visual experts. This approach is more sophisticated than merely aggregating extra data or parameters.
> > - **Combination of Verification Scores from Small Visual Experts and Self-Correctness of LLMs**: Although the step-by-step self-correctness ability of LLMs has already demonstrated significant potential [3, 4], our work is the first to elegantly combine these verification scores with self-correctness, which manages to strike a balance between efficiency and efficacy.
> >
> >
> > [1] Multimodal Explanations by Predicting Counterfactuality in Videos
> >
> > [2] Towards Counterfactual Image Manipulation via CLIP
> >
> > [3] Let's Verify Step by Step
> >
> > [4] Automatically Correcting Large Language Models: Surveying the landscape of diverse self-correction strategies

---

> ### Author Response · Authors · 2023-11-15
> **Looking forward to your feedback and discussion**
>
> Dear Reviewer 2ndj,
>
> We thank you again for your valuable comments and suggestions to help us improve our paper in the aspects of presentation, experiments, analysis, and discussion. We also appreciate that you noticed our `innovative concept in the field of visual programming`, `the robustness of the proposed methodology is evident through its application to two distinct models, yielding a notable 4% improvement in accuracy for compositional visual question answering tasks`, `a thorough preliminary study, demonstrating how individual components contribute to enhanced performance`, `excels in clearly articulating its methodology`,  `detailed explanations`, `the contributions of this paper are **highly significant** for the field of visual compositional reasoning`,  ` **addresses critical limitations** in existing methodologies and offers a **substantial improvement** in prediction accuracy, marking a notable advancement in the field`. These compliments have encouraged us a lot and will help us to make a timely contribution and broader impact on both the Computer Vision and the compositional reasoning community.
>
>
> Our current revised paper has taken into account all your suggestions, please kindly let us know if you still have follow-up questions.
>
> Thanks,
>
> Authors

---

> ### Comment · Reviewer_2ndj · 2023-11-22
> **Official Comment from Reviewer 2ndj**
>
> Hi there,
>
> Thanks for addressing all of my questions. I appreciate your efforts in conducting additional error analysis on the failure cases. It makes sense to me that extra errors would be introduced when aggregating the modules, and the performance could be bottlenecked by one particular one (in your case, seems to be the VQA model). I've updated the soundness score in recognition of your analysis.

---

> > ### Author Response · Authors · 2023-11-23
> > **Thank you for the Reply**
> >
> > Dear 2ndj
> >
> > Thank you for reading our rebuttal and confirming that **all questions are addressed**. We are happy to know that our additional error analysis on the failure cases is helpful and we will definitely put them in the final revision. Thank you also for updating the soundness score to **4 excellent**, which definitely encourages us and acknowledges the integrity and soundness of our work. If the discussion time is allowed, we are also happy to answer if you still have any other concerns about the overall rating.
> >
> > Best,
> >
> > Authors

---

### Official Review · Reviewer_Tiuv · 2023-11-10

**Soundness:** 3 good
**Presentation:** 3 good
**Contribution:** 3 good
**Rating:** 6
**Confidence:** 3

**Summary:**

This work focuses on resolving two key classes of errors in existing visual programming approaches:
1. Reasoning Error: Error in common sense reasoning regarding which operation to perform
2. Module Error: Error by individual modules (like object detection) in their outputs.

They introduce ExoViP, which adds verification steps (Image-text matching, captioning and VQA verifier) for each module to detect errors before they propagate. To solve planning errors, they maintain multiple reasoning traces and use beam search to select the best sequence of neural modules. The authors evaluate the method on five visual composition reasoning tasks (Compositional image question answering, Referring Expression Segmentation, Natural Language Visual Reasoning, Abstract Reasoning, and Language guided Image Genereation).

**Strengths:**

1. The authors propose a well-motivated idea to overcome module-based and planning-based errors in Visual Programming through the use of verification scores and trace reasoning.
2. The method reuses existing modules for obtaining verification scores.
3. The evaluation spans multiple common Visual Composition Reasoning problems with an analysis for each type of problem.
4. The writing is good, and the paper is easy to follow

**Weaknesses:**

1. The initial analysis finds the percentage of errors caused by the two types of problems (module error and planning error). The developed method's intuition is to solve these two errors with the two mentioned strategies. However, the final analysis does not include the kind of errors this method actually ends up fixing and in what ratio. It would be beneficial to show the statistics that demonstrate what proportion of errors of each type were fixed by the proposed methods compared to VISPROG [1].
2. The method is not evaluated with any open-source Large Language Models (LLaMA, Vicuna, etc.) Restricting evaluation only to GPT-3.5-turbo reduces the generalizability of the approach.
3. The performance increment across tasks over VISPROG [1] is still incremental, considering the time cost of using beam search and multiple verification modules.

**Questions:**

1. In section 4.2 - You sort the candidate neural modules according to verification scores. From what your paper says verification scores are obtained for different candidates generated by a single neural module. Are these verification scores for the neural modules themselves different or are they the verification score of the top candidate of each neural module?
2. What is the inference latency for your approach compared to using VISPROG [1]? The use of beam search and multiple verification modules indicates a significant added overhead.
3. Both ViperGPT [2] and VISPROG [1] implementations used GPT-3 as the model of choice. Will the advantages of your method transfer to GPT-3 and other Open LLMs as well? Or is GPT-3.5-turbo necessary in order to accomplish Self-Correction and Trace Reasoning?
4. Could you report an analysis about what percentage of module errors and planning errors were fixed by your method compared to VISPROG?

[1] Tanmay Gupta and Aniruddha Kembhavi. Visual programming: Compositional visual reasoning without training. In Conference on Computer Vision and Pattern Recognition (CVPR), pp. 14953– 14962, June 2023

[2] Sur\'is D\'idac, Sachit Menon and Carl Vondrick, Sachit Menon, and Carl Vondrick. Vipergpt: Visual inference via python execution for reasoning. arXiv preprint arXiv:2303.08128, 2023.

---

> ### Author Response · Authors · 2023-11-14
> **Response to Reviewer Tiuv (1/2)**
>
> Thank you for your constructive comments and acknowledgment of our well-motivated idea. We carefully address each of your concerns as follows.
>
>
> > W1: The initial analysis finds the percentage of errors caused by the two types of problems (module error and planning error). The developed method's intuition is to solve these two errors with the two mentioned strategies. However, the final analysis does not include the kind of errors this method actually ends up fixing and in what ratio. It would be beneficial to show the statistics that demonstrate what proportion of errors of each type were fixed by the proposed methods compared to VISPROG [1].
>
> > Q4: Could you report an analysis about what percentage of module errors and planning errors were fixed by your method compared to VISPROG?
>
> **A:** Thank you for the valuable suggestion. In our original submission, we have manually analyzed 100 randomly sampled failure cases on VisProg. We find that there are three typical reasons for the failures: a) vision module prediction error; b) LLM planning error; c) others. We demonstrate the statistics of the failure cases in **Appendix.C**. We provide illustrations of two representative failure cases resulting from our methodology in **Appendix I**.
>
> In our revised paper, following the application of our proposed framework, we re-assessed the same cases. As shown in **Appendix C**, "`we were pleased to discover a reduction in module errors by 28.87%, and a decrease in planning errors by 42.35%. Nevertheless, our current strategy was unable to rectify 69.8% of the errors. When juxtaposed with the data from Table 1, our method has enhanced VISPROG by 7.11%, which is lower than the improvement of the failure cases. This outcome suggests that our approach may give rise to novel challenges. We further demonstrate common errors of our method in Figure 10 and Figure 11, As highlighted in the study of failure cases, the majority of these instances originate from the VQA module.`"
>
> > W2: The method is not evaluated with any open-source Large Language Models (LLaMA, Vicuna, etc.) Restricting evaluation only to GPT-3.5-turbo reduces the generalizability of the approach.
>
> > Q3: Both ViperGPT [2] and VISPROG [1] implementations used GPT-3 as the model of choice. Will the advantages of your method transfer to GPT-3 and other Open LLMs as well? Or is GPT-3.5-turbo necessary in order to accomplish Self-Correction and Trace Reasoning?
>
> **A:** Thank you for the helpful suggestion. In fact, both the official implementations of [ViperGPT](https://github.com/cvlab-columbia/viper) and [VISPROG](https://github.com/allenai/visprog) are supported by the GPT-3.5-turbo. In accordance with this line of research, all the experimental results reported herein are also built on the GPT-3.5-turbo model. In addition, we replaced the GPT-3.5-turbo with LLama2-chat-13b and obtained the following results. We are thrilled to discover that our method can yield significant improvements in open LLM. We have added these results to **Section 5.3** in our revised paper.
>
>
> | model | accuracy on GQA |
> | -------- | -------- |
> | VISPROG (GPT-3.5-turbo)     | 57.41     |
> | ExoViP (GPT-3.5-turbo)     | 61.49     |
> | VISPROG (Llama-2-13b-chat)    | 46.41     |
> | ExoViP (Llama-2-13b-chat)     | 54.45     |
>
>
> > W3: The performance increment across tasks over VISPROG [1] is still incremental, considering the time cost of using beam search and multiple verification modules.
>
> > Q2: What is the inference latency for your approach compared to using VISPROG [1]? The use of beam search and multiple verification modules indicates a significant added overhead.
>
> **A:** Thank you for the constructive advice. We present the mean inference time on the GQA dataset. Generally, the temporal expenditure of our tree-based search method significantly surpasses that of VisProg. However, the majority of the time is consumed by the call of the  OPENAI API, an issue we posit is intrinsic to analogous works [3,4,5]. When compared to Depth First Search/ Breadth First Search [3] or Monte Carlo Tree Search [4,5], we assert that our beam search-based method can achieve an optimal equilibrium between efficiency and effectiveness. We have added this part to **Appendix D** in our revised paper.
>
> | model   | Total Inference Time (s) |  Planning Time (s)   | Module Inference Time (s) |
> | ------- | -------------------- | --- | -------------------------- |
> | VisProg | 1.59                 |  1.10   | 0.49    |
> | ExoViP  | 4.32                 |   3.64  | 0.68         |
>
>
>
>
>
>
> [1] Visual Programming: Compositional Visual Reasoning Without Training
>
> [2] ViperGPT: Visual Inference via Python Execution for Reasoning
>
> [3] Tree of Thoughts: Deliberate Problem Solving with Large Language Models
>
> [4] Alphazero-like Tree-Search can Guide Large Language Model Decoding and Training
>
> [5] Language Agent Tree Search Unifies Reasoning Acting and Planning in Language Models

---

> > ### Author Response · Authors · 2023-11-14
> > **Response to Reviewer Tiuv (2/2)**
> >
> > > Q1: In section 4.2 - You sort the candidate neural modules according to verification scores. From what your paper says verification scores are obtained for different candidates generated by a single neural module. Are these verification scores for the neural modules themselves different or are they the verification score of the top candidate of each neural module?
> >
> > **A:** Sorry about the confusion. In the tree-based search algorithm, the verification score for a single node is calculated as the average score of the top candidate. The accumulated score of the candidate path is then used for sorting purposes. We have clarified this part in **Section 4.2** of our revised paper.

---

> ### Author Response · Authors · 2023-11-15
> **Looking forward to your feedback**
>
> Dear Reviewer Tiuv,
>
> Please kindly let us know if you still need further clarification. We will be more than happy to answer.
>
> We thank you again for your valuable comments and suggestions to help us improve our paper. We have tried our best to address all your concerns. We have also substantially revised our paper by following other valuable suggestions from other reviewers. Please take some time to look at them and see if you have further questions.
>
> Best,
>
> Authors

---

> ### Comment · Reviewer_Tiuv · 2023-11-22
>
> Thank you for the clarifications and the detailed response. I believe adding error analysis of your method on the selected cases and results on open source LLMs warrants an improved soundness score (I have updated the soundness score accordingly). I will maintain my overall rating as positive feedback.

---

> > ### Author Response · Authors · 2023-11-23
> > **Thank you for Reply**
> >
> > Dear Reviewer Tiuv,
> >
> > Thank you for reading our rebuttal and confirming that our analysis `warrants an improved soundness score`. Thank you for updating the soundness score as well.
> >
> > Authors

---

### Author Response · Authors · 2023-11-14
**Reseponse to All Reviewers**

Dear reviewers,

We appreciate the efforts of all reviewers and your constructive suggestions! We highlight the main strengths and contributions of our work as follows:

- **Contributory Idea**. We propose "a well-motivated idea to overcome module-based and planning-based errors in Visual Programming" (Reviewer **Tiuv**) through the use of "innovative concept" (Reviewer **2ndj**) of verification modules and trace reasoning, which is "worth diving into this area" (Reviewer **atdm**) to mitigate these two types of errors. "The contributions of this paper are highly significant for the field of visual compositional reasoning" (Reviewer **2ndj**)
- **Effective Method**. Our method is "reasonable" (Reviewer **rVcU**) and could "effectively tackles both module and planning errors", and "the robustness of the proposed methodology is evident" (Reviewer **2ndj**)
- **Comprehensive Experiments**. Our work "conducts a thorough preliminary study, demonstrating how individual components (such as self-correctness, beam search, and verification) contribute to enhanced performance", In addition, our method "yields a notable 4% improvement in accuracy for compositional visual question answering tasks" (Reviewer **2ndj**). Moreover, empirical results "exhibit consistent enhancement across five compositional reasoning tasks". (Reviewer **rVcU**)

We've revised our manuscript per the reviewers' suggestion (highlighted in red in the uploaded revision pdf). Detailed responses to each reviewer's concerns are carefully addressed point-by-point. Below summarize the major updates we've made:

- clarify the verification score and reasoning trace-searching algorithm. (**Section 4.2**)
- add failure cases analysis before and after applying our method. (**Appendix C**)
- add results of our method applying on open LLMs. (**Section 5.3**)
- add time efficiency study of our method. (**Appendix D**)

We believe our proposed method will make a timely contribution to both Computer Vision and compositional reasoning community. Below we provided detailed responses to address each reviewer's concern and are more than willing to be involved in further discussions if there are further comments.

Best,

Authors

---

### Meta-Review · Area_Chair_orii · 2023-12-15

**Metareview:**

This paper introduces ExoViP, a novel approach for enhancing compositional visual reasoning. Its strengths lie in the novel concept of employing verification modules to tackle both module and planning errors in visual programming. The paper demonstrates robust empirical evaluation, with comprehensive experiments showing substantial improvement in accuracy across several compositional reasoning tasks.

Despite these strengths, the paper has several critical limitations. As reviewers pointed out, the development and application of verifier modules lack sufficient depth, especially in their selection process and effectiveness, with no extensive ablation studies conducted. The evaluation's focus on the VISPROG dataset limits the generalizability of the findings, and the comparison with inconsistent baselines is less convincing.

Overall, while the innovative approach and empirical evidence of ExoViP's efficacy in improving compositional visual reasoning tasks are acknowledged, the methodological gaps, concerns about generalizability, and insufficient error analysis lead to a rejection for this work. Further development and refinement of ExoViP could potentially address these issues and make it a more solid and impactful work.

**Justification For Why Not Higher Score:**

Same as above.

**Justification For Why Not Lower Score:**

N/A

---

### Decision · Program_Chairs · 2024-01-16

Reject